# Potential Use of Thalidomide in Glioblastoma Treatment: An Updated Brief Overview

**DOI:** 10.3390/metabo13040543

**Published:** 2023-04-11

**Authors:** Ahmed Ismail Eatmann, Esraa Hamouda, Heba Hamouda, Hossam Khaled Farouk, Afnan W. M. Jobran, Abdallah A. Omar, Alyaa Khaled Madeeh, Nada Mostafa Al-dardery, Salma Elnoamany, Eman Gamal Abd-Elnasser, Abdullah Muhammed Koraiem, Alhassan Ali Ahmed, Mohamed Abouzid, Marta Karaźniewicz-Łada

**Affiliations:** 1Department of Cell Biophysics, Faculty of Biochemistry, Biophysics and Biotechnology, Jagiellonian University in Kraków, 31-007 Kraków, Poland; 2Faculty of Medicine, Menoufia University, Menoufia P.O. Box 5744, Egypt; 3Faculty of Medicine, Fayoum University, Fayoum P.O. Box 63514, Egypt; 4Faculty of Medicine, Al Quds University, Jerusalem P.O. Box 51000, Palestine; 5Department of Pharmaceutical Services and Sciences, Children’s Cancer Hospital Egypt (CCHE-57357), Cairo 11617, Egypt; 6Faculty of Medicine, Alexandria University, Alexandria 21311, Egypt; 7Faculty of Pharmacy, Al-Azhar University, Naser City, Cairo 11884, Egypt; 8Department of Bioinformatics and Computational Biology, Poznan University of Medical Sciences, 60-812 Poznan, Poland; 9Doctoral School, Poznan University of Medical Sciences, 60-812 Poznan, Poland; 10Department of Physical Pharmacy and Pharmacokinetics, Poznan University of Medical Sciences, Rokietnicka 3 St., 60-806 Poznan, Poland

**Keywords:** glioblastoma, thalidomide, antineoplastic agents, angiogenesis, chemotherapy, VEGF, TNF

## Abstract

Glioblastoma is the most common malignant primary brain tumor in adults. Thalidomide is a vascular endothelial growth factor inhibitor that demonstrates antiangiogenic activity, and may provide additive or synergistic anti-tumor effects when co-administered with other antiangiogenic medications. This study is a comprehensive review that highlights the potential benefits of using thalidomide, in combination with other medications, to treat glioblastoma and its associated inflammatory conditions. Additionally, the review examines the mechanism of action of thalidomide in different types of tumors, which may be beneficial in treating glioblastoma. To our knowledge, a similar study has not been conducted. We found that thalidomide, when used in combination with other medications, has been shown to produce better outcomes in several conditions or symptoms, such as myelodysplastic syndromes, multiple myeloma, Crohn’s disease, colorectal cancer, renal failure carcinoma, breast cancer, glioblastoma, and hepatocellular carcinoma. However, challenges may persist for newly diagnosed or previously treated patients, with moderate side effects being reported, particularly with the various mechanisms of action observed for thalidomide. Therefore, thalidomide, used alone, may not receive significant attention for use in treating glioblastoma in the future. Conducting further research by replicating current studies that show improved outcomes when thalidomide is combined with other medications, using larger sample sizes, different demographic groups and ethnicities, and implementing enhanced therapeutic protocol management, may benefit these patients. A meta-analysis of the combinations of thalidomide with other medications in treating glioblastoma is also needed to investigate its potential benefits further.

## 1. Introduction

Glioblastoma is adults’ most common malignant primary brain tumor [1]. Maximal safe surgical resection followed by radiotherapy and/or chemotherapy is the glioblastoma standard treatment protocol [2]; however, the median survival is short, only about 14 to 16 months, and a 5-year overall survival rate of less than 10% [3]. Glioblastomas are considered solid tumors. They are composed of heterogeneous cancer cells that create new blood vasculatures, inflammatory elements, and stromal environments [4,5,6,7]. The most important factor in diagnosing glioblastoma is the detection of isocitrate dehydrogenase (*IDH*) mutations. According to that, the 2021 Fifth Edition World Health Organization Classification of central nervous system (CNS) tumors has classified glioblastoma into two groups: IDH–wild-type glioblastoma (hereinafter referred to as GBM for simplicity) (95%) and IDH–mutant astrocytoma (5%) [2]. GBM accounts for 45.2% of primary malignant brain and CNS tumors, with a yearly incidence rate of 3.19 per 100,000 people [1].

During recurrent GBM, chemotherapy remains the most effective approach, because patients cannot redo surgeries or irradiation [8]. Vascular endothelial growth factor (VEGF) is a prominent mediator of tumor angiogenesis [9,10], and there is increasing scientific interest in studying VEGF inhibitors. Bevacizumab was the first antiangiogenic drug to treat GBM, approved by the US Food and Drug Administration (FDA). Bevacizumab is used as a single medication in patients with GBM who have progressive disease following front-line therapy that consists of surgical resection, radiotherapy, and temozolomide [6,11,12,13]. The tumor-initiating cells (TICs) and glioblastoma stem cells (GSCs) play an essential role in GBM development, via different processes starting at the subventricular zone along ventricles underlying driver mutations. Both cell types have stem cell characteristics such as self-renewal, differentiation, and a low rate of proliferation. These features of GSCs contribute to chemotherapy resistance by increasing the drug efflux pump, drug degradation, DNA repair, and decreasing the drug influx pump, the activation of prodrugs. For example, aldehyde dehydrogenase 1 (*ALDH*) and O6-methylguanine-DNA methyltransferase (*MGMT*) are overexpressed in GBM, and decrease the activity of alkylating agents. The latter has a critical role in DNA repair, so the DNA methylation of the *MGMT* gene is associated with a better outcome and a high survival rate. Hypoxia is considered to be one of the characteristics of GBM; therefore, the GSCs can resist chemo- and radiotherapy through mitochondria modulation, which controls apoptosis and tumor cells’ necrosis [14,15].

Wang et al. showed that bevacizumab compared to thalidomide (THD) for recurrent GBM patients, improved the objective response rate (ORR) and 6-month median progression-free survival, but not the 1-year median overall survival [16]. Therefore, many clinical studies test new angiogenesis inhibitors to enhance GBM treatment results [17]. Moreover, highlighting different mechanisms of THD in vivo and in vitro will provide a higher chance of understanding cell behavior and resistance to anticancer therapies [18,19].

We conducted an exhaustive review of the literature, and aim to highlight the THD mechanism of action in vivo and in vitro and its usefulness when co-administered with other VEGF inhibitors to treat GBM. To the best of our knowledge, no similar study has been performed.

## 2. Genetic and Molecular Pathogenesis of GBM

The genetic features of IDH–wild-type glioblastoma include mutation in the telomerase reverse transcriptase (TERT) promoter; amplification of epidermal growth factor receptor (EGFR); loss of heterozygosity for 10q; deletion and point mutation of phosphatase and tensin homolog (PTEN); methylation of the MGMT promoter; a mutation in BRAF V600E. The features of IDH-mutant glioblastoma include mutation of ATRX (alpha-thalassemia/mental retardation, X-linked), TP53, DH1/IDH2, and PDGFRA (platelet-derived growth factor receptor alpha) amplification, and loss of heterozygosity for 10q and 19q [2]. According to primary and secondary GBM, primary GBM has a mutation in PTEN (MMAC1), amplified EGFR, and chromosome 10 loss [20,21]. On the contrary, mutant IDH is more likely to occur in secondary GBM [22]. The IDH mutation is used as a marker for secondary GBM, and frequently occurs in more than 80% of cases; however, it rarely occurs in primary GBM, in around 5% of cases [23,24,25,26]. The amplification of EGFR in secondary GBM is rare, which tends to have TP53 mutations [27]. The differences in genetic and epigenetic profiles of primary and secondary GBM are not shown in histological examinations as different characteristics, even though they are believed to share different cell origins. Surprisingly, cells with mutated IDH1 and IDH2 tend to have better prognoses [28]. Methylation of MGMT prolongs survival compared to the unmethylated form of MGMT, which is involved in deoxyribonucleic acid (DNA) repair upon administrating alkylating treatment [29]. Myeloid-derived suppressor cells (MDSCs) play an essential role in decreasing the activity of the T-cell response in glioma, and it is an immature heterogeneous group of cells recognized for their myeloid lineage [30]. MDSCs weaken the first line of defense by inhibiting the natural killer cell activation receptor (NKG2D), and decreases interferon-gamma (IFNγ) production while transforming growth factor-beta (TGFβ) is present [31]. In addition to that, MDSCs also diminish adaptive immunity by disturbing the production of ARG1, inducible nitric oxide (NO) synthase 2 (iNOS2), TGFβ, depletion of cysteine, and the down-regulation of CD62L (L-selectin) [32,33]. Angiogenesis and vasculogenesis in the solid tumor are correlated with a poor prognosis, due to their promotion by MDSCs [34].

## 3. Hypoxia, Endoplasmic Reticulum Stress, and Chemoresistance

GBM is characterized by hypoxic regions that are responsible for necrosis associated with GBM. The low-oxygen condition could activate the hypoxia-inducible factors (HIFs), critical transcriptional factors that play an essential role in a tumor’s response against hypoxia [35,36]. HIFs are believed to be sensors and proteins that are released in hypoxic regions responsible for cell adaption to low levels of oxygen, by optimizing the cellular processes that could initiate transcription factors activation of specific genes responsible for angiogenesis, metabolic reprogramming, and chemo-radio resistance in GBM [37,38,39,40,41]. The PTEN mutation (20–40%) and P53 mutation have been found in GBM. A loss of their function leads to the overexpression of HIF-1α. Moreover, HIF-1α can increase VEGF levels as one of the angiogenesis proteins. The level of VEGF in the cyst fluid of glioblastoma patients was found to be 200–300 fold more than in the serum.

During hypoxia, HIF-1α induces anaerobic glycolysis in GBM, resulting in a reduction in mitochondrial respiration, an increase in lactate levels and tumor acidity, and an interruption in the pH ratio between the intracellular and extracellular matrix that could decrease the passive absorption for many drugs, therefore increasing the probability of drug resistance [42,43]. Several studies reported that as the tumor size increases, the pH decreases. An acidic pH also plays an essential role in regulating cell proliferation, angiogenesis, immunosuppression, invasion, and chemoresistance in solid tumors [44,45,46,47,48]. Vaupel et al. reported that the pH could vary among the localized region of the tumor [48]. Normal brain tissue has a pH of 7.1, and in brain tumors it can be 5.9 [49]. Interestingly, upon performing electrode measurements, Hjelmeland et al. [49] reported a significant reduction in pH at the edge of the tumor compared with that in normal tissues. Moreover, the pH at the center of the tumor was even lower. Such a shift in the pH in gliomas, and the low pH, may increase angiogenesis through the induction of VEGF [50,51]. Acidic stress in GSCs promotes the HIF-2α protein. In low pH, HIF-2α mRNA increased 7-fold in cultures used for GSCs; however, HIF-1α mRNA was repressed. These findings hypothesize that HIF-2α could be responsible for inducing VEGF in acidic conditions [49]. Notably, the stability of HIFs is known to be regulated at the protein level as well, since HIFs have been found to be stable at low pH [52].

Other involvements of HIF-2α in GBM have also been reported. Li et al. [53] highlighted a higher expression level of HIF-2α in GSCs than in non-GSC. However, minimal expression of HIF-2α was found in normal adult murine neural progenitors under both conditions of hypoxia, and normoxia. Interestingly, HIF-1α was not as dramatically upregulated as HIF-2α in response to hypoxia in GSCs. The HIF-2α mRNA half-life time was found to be shorter in GSCs in comparison with non-GSCS. This indicates an increased de novo synthesis of mRNA of HIF-2α rather than the stabilization of mRNA of HIF-2α [53]. VEGF promoter activity was severely affected by the knock-down of HIF-1α and HIF-2α in hypoxic conditions, and decreased VEGF mRNA levels and intracellular and secreted VEGF protein levels. In contrast to non-stem glioma cells, only HIF-1α was required to regulate VEGF [53]. In another study, where the mesenchymal shift is the process where cells lose their adhesion and become migratory and invasive, it was found that HIF-1α-ZEB1, not the HIF-2α signaling axis, promoted this feature in GBM [54]. Another protein was associated with HIF-2α is teneurin transmembrane protein 1 (TENM1), a family of transmembrane proteins located on chromosome X. In vertebrates, their expression occurs during the central nervous system development, and is also associated with cellular signaling, cell proliferation, and adhesion regulations [55,56]. A recent study also reported that HIF-2α silencing decreased the expression of TENM1 [57]. Cancer stem cells show increased expressions of CD133 and HIF-2α when they are exposed to hypoxia [53,58,59,60].

As a result of hypoxia, endoplasmic reticulum (ER) stress occurs due to the accumulation of unfolded proteins and misfolded proteins because of the shortage of energy produced by the aerobic process; as the ERAD/ERAQ system does not work properly, it is becomes overburdened [61,62,63]. Consequently, it drives the cancer cell population in a similar fashion of selecting and allowing for the survival of the most resilient phenotype of glioma cells that can withstand hypoxic and endoplasmic reticulum stress, and are accustomed to resisting anti-tumor drugs [61,64]. The unregular and unorganized structure of solid tumor vasculature contributes to irregular drug delivery to the site of action, whereas the well-oxygenated regions are less susceptible to ER stress [65,66]. HIF-1α is associated with the multidrug resistance mutation 1 (*MDR1*) gene that is responsible for the expression of P-glycoprotein/ATP binding cassette transporter B1 [Pgp/ABCB1] [67]. Upon studying doxorubicin resistance, a direct link was found between HIF-1α and increased expression of P-glycoprotein (Pgp), which effluxes the medication outside the cell. [68]. It was observed that the efficacy of temozolomide was increased after the knockdown of HIF-1α, due to the down-regulation of DNA repair proteins [69,70].

## 4. Thalidomide Mechanism of Action

The mechanism of action of THD has not been clearly defined or explained. Nevertheless, several studies suggest that THD is associated with immunity, inflammation modulators, and angiogenesis properties in tissues. THD actions have different complex effects on cytokines [71,72,73]. Tumor necrosis factor-alpha (TNFα), IFNγ, cyclooxygenase 2 (COX2), and interleukins (ILs) IL-10 and IL-12 are among the inflammatory cytokines that THD effects [74]. THD primarily targets cereblon (CRBN), which decreases the release of cytokines TNFα and interleukins through degrading TNFα mRNA [71,72,75,76,77,78]. CRBN is a protein whose gene is found in human chromosome 3 [79]. CRBN’s role has not been clearly explained, although it is connected with ubiquitin ligase that binds to the damaged DNA binding protein (DDB1), Cullin-4A (CUL4A), and its regulators [80]. In addition, CRBN has also been associated with the large-conductance calcium-activated potassium channel (KCNMA1) [81,82].

It is suggested that THD binding to CRBN limits angiogenesis ability, and helps generate reactive oxygen species [78,83]. THD induces T helper 2 (Th2) production; on the contrary, it inhibits the Th1 release in peripheral mononuclear cells [71] (Figure 1).

TNFα down-regulation causes nuclear factor kappa B (NF-kB) inhibition that results in lower levels of interleukin-6 transcriptions. Oppositely, THD up-regulates the caspase-8 protein widely used in myeloma by inducing caspase-8 myeloma cell programmed cell death apoptosis [84]. However, THD’s entire mechanism of action in reducing myeloma cells is still unclear. THD stimulates natural killer cells and T-lymphocytes, and prevents strong myeloma cells’ adhesion to bone marrow [85], thus affecting the composition of the bone marrow microenvironment [86]. THD has an antiangiogenic effect by blocking growth factors such as VEGFs and FGFβ [74,87,88,89,90]. Other studies have shown that THD inhibits HOXB7 [91] and Sp1 that have a binding site in the c-MYC promoter and suppress the TGFβ1-mediated non-SMAD ERK1/2 signaling pathways [92]. Several uses and mechanisms of actions in some clinical conditions exist in (Table 1) [93,94,95,96,97,98,99,100,101,102,103,104,105,106,107,108,109,110,111,112,113,114,115,116,117].

## 5. Thalidomide Efficacy

It has been over sixty years since THD has been on the market, and it is still frequently used in a wide range of therapeutic applications. The clinical trials and pharmacovigilance research have shown that THD is an effective medication in the treatment of idiopathic pulmonary fibrosis (IPF), severe lung injuries caused by swine flu subtype H1N, and lung injuries caused by the toxic fast-acting herbicide called paraquat, with a well-defined mode of action [118]. To determine THD efficacy, we performed a systematic search on various databases, and reported the results of clinical trials that used THD for GBM treatment (Figure 2).

It has been shown that THD may be a promising therapeutic option for treating GBM through the following three trials conducted on patients with recurrent GBM who received conventional chemotherapy followed by THD [120,121,122]. During the trials, the daily dose of THD was between 100–1200 mg, with good tolerance in general, although there were uncommon responses. The first study included patients with recurrent GBM (n = 18), 77.8%. The median overall survival (OS) for 17 patients was 36 weeks (12–40) [123]. The second trial included patients with recurrent GBM (n = 42), 38 of whom were eligible for assessment; only 42% had stable disease, and 5% had a partial response. The OS was 31 weeks, and 35% of the patients achieved one-year survival [124]. The third trial included patients with recurrent GBM (n = 39), of whom 36 were eligible for assessment; only 33% had stable disease, 6% had a partial response, the OS was 28 weeks, and eight patients achieved one-year survival [125].

The use of THD concurrently with irinotecan has limited efficacy for the treatment of newly diagnosed or recurrent GBM; this was proven through two studies. In the first study (n = 26), 24 were eligible for assessment; only 79% had stable disease and 7% had a partial response. The six-month progression-free survival (PFS) of the recurrent group was 19%, and the six-months PFS of the newly diagnosed group was 40% [126]. In the second study, patients with recurrent GBM only (n = 32) were eligible for assessment. Only 19 patients had stable disease, one had a partial response, one had a complete response, the one-year OS was 34%, and the six-month PFS was 25% [127]. It is worth mentioning that THD was reported in eight studies to be an ineffective drug for the treatment of GBM [87,128,129,130,131,132,133].

A systematic review and meta-analysis compared only two cohorts of THD (n = 81) to 7 cohorts of bevacizumab (n = 351). The ORR favored bevacizumab over THD (RR 6.8, 95%CI 2.64–17.6; *p* < 0.001); however, both drugs showed comparable results in the progression-free survival and 1-year median overall survival rates (RR 1.68, 95%CI: 0.84–3.34, *p* = 0.07; and RR 0.89, 95%CI: 0.59–1.37; *p* = 0.31, respectively) [16]. In addition to using THD in the direct treatment of GBM, Hassler et al. [134] suggested using THD as a palliative treatment in patients with advanced secondary GBM. The patients (n = 23) were advised to administer 100 mg of THD at bedtime. Symptomatic improvement was most prominent in restoring a normal sleep pattern, and 47.8% of the patients survived longer than one year [134]. The palliative effects of THD can be a topic for further research. A list of the combined possible uses of THD with other drugs is shown in Table 2 [74,85,86,87,88,89,90,91,92,135,136,137,138,139,140,141,142,143,144,145,146,147,148,149,150,151,152,153,154,155].

## 6. Thalidomide Safety

The proven teratogenicity of THD made it strictly prohibited from being used during pregnancy. Before THD may be taken, all patients have to be enrolled in the System for THD Education and Prescribing Safety (STEPS) program. Prior to commencing treatment with THD, women of reproductive potentiality (less than two years postmenopausal) are required to show a negative pregnancy test. In addition, they have to use two different effective ways of birth control and be examined for pregnancy every four weeks; meanwhile, men who are going to take THD have to quit any sexual activities, or use condoms made of latex [121]. THD is known to cause many side effects, including numbness, nervousness, confusion, paresthesia, aural buzzing, nausea, up to 20%, increased appetite, 30% of burning sensation and deep vein thrombosis, 5% of bradycardia and neutropenia, 25% of skin rash, hangover feeling, decreased libido, edema, and hypothyroidism up to 25% [121]. THD’s common side effect was initially marketed as a sedative drug. The intensity of sedation tends to lessen with prolonged usage at a steady dose, which could be decreased and avoided by taking medicine at night three hours before bed. Constipation is a typical side effect that can vary depending on the dosage. Patients should be encouraged to eat food that contains a high amount of fiber, and to take laxatives as needed. In the case of dry skin side effects, it is pretty common to see it with pruritus, with the severity based on the drug’s dosage. Nevertheless, this effect may be reduced by using lubricants that are free of alcohol [83]

## 7. Thalidomide Toxicity

THD is known for its considerable risk to the unborn and developing child. As a result, THD and its derivatives are highly controlled, and necessitate the use of contraception when used as a therapy. Patients should cease using THD if they have serious side effects, especially neuropathic symptoms [156]. THD toxicity in the treatment protocol for patients with GBM was studied, and a trial was conducted on 39 patients; 26 received the full dosage of the THD regime. The results showed that the majority of patients tolerated THD well. However, four incidences were graded as 4 for cortical toxicity, and were all seizures. All of the participants in this experiment had a previous seizure history, and it was verified that tumor development increased during the activity of new seizures. Additionally, one incident involved constipation grade 1, and there were six incidences of grade 2 constipation episodes. Moreover, twelve patients had somnolence grade 1, three had grade 2, and six had grade 3. These were the most prevalent toxicities linked to THD [157]. A further phase II trial was conducted on 17 GBM patients to assess THD toxicity, and it was found that a few cases were graded. Two patients (17%) showed leukopenia progression grade 3–4 with neutropenia. Furthermore, other serious adverse events comprised (4 patients, 24%) those with fatigue grade 3. Moreover, another patient suffered from neurotoxicity grade 3. In this intervention sample, the medication-related toxicity had no clinical consequences, and no participant was discontinued from the experiment due to THD toxicity (Figure 3) [125].

## 8. Thalidomide Drug Interactions

THD is mainly eliminated from the body by a hydrolysis reaction that occurs across the distribution area of the drug, and there is some influence on hepatic elimination and excretion via the kidney. The metabolization of THD in our bodies is minimally done by CYP450, and 2C19 is engaged in the synthesis process of 5-OH metabolites [160]. In liver microsomes, THD was found to suppress CYP2C19 function while increasing the activity of CYP3A, which means the elevation of CYP3A is due to a heterotopic interactivity rise in CYP3A5. In animal trials, THD revealed a possibility of human CYP3A up-regulation with a greater midazolam clearance with midazolam usage as a pharmacological probe. Based on a recent study, THD acts as a ligand for the pregnane X receptor (PXR) and the constitutive androstane receptor CAR, increasing the activity of the CYP450 enzyme. Since in vitro studies indicated that THD needs CYP450 metabolic stimulation, and that metabolites are detectable in urine, weak metabolizers of CYP2C19 (about 15% of Asians and 25% of Caucasians) may need greater THD dosages. Extensive metabolizers may be at a high risk of side effects. As a result, the CYP2C19 genotype and the CYP2C19 inhibitors or inducers could influence exposure to the active metabolite of THD [161]. Furthermore, adverse effects of THD on the central nervous system CNS, including dizziness, sleepiness, and focusing problems, may be exacerbated by alcohol intake while using THD. Some patients may also have difficulty concentrating and making decisions. While using THD, patients must abstain from or minimize their alcohol use. They should not exceed the prescribed dosage, and refrain from engaging in tasks requiring wakefulness, such as driving or working with dangerous equipment [162]. Figure 4 represents some of adverse effects of THD drug interactions.

## 9. Conclusions

GBM is still one of the most life-threatening types of primary malignant brain tumors, with a poor prognosis and a low 5-year survival rate. Investigating the different mechanisms of action of THD could lead to new potential uses for the drug, especially when combined with other medications. Future research on treatment options should also consider existing drugs that have different mechanisms for fighting tumors. Additionally, more efforts are needed to develop accurate preclinical models to evaluate treatments’ effectiveness and improve methods for the early diagnosis of GBM.

## Figures and Tables

**Figure 1 metabolites-13-00543-f001:**
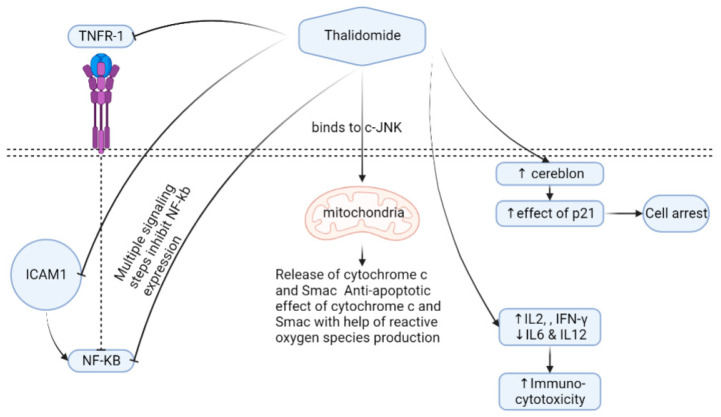
Thalidomide mechanism of action. Cells expressing the cereblon protein show higher expressions of p21 if thalidomide is administrated, leading to cell arrest. Thalidomide can also increase immuno-cytotoxicity through the regulation of different cytokines. Thalidomide has a direct effect on mitochondria through binding to c-JNK, followed by releasing oxygen species. The inhibition of NF-KB occurs as a means for thalidomide by interacting with TNFR-1, ICOM1, and other multistep signaling. Abbreviations, c-JNK—c-jun terminal kinase; TNFR1—tumor necrosis factor receptor 1; IFN—interferon, IL—interleukin; NF-kB—nuclear factor-kappa B; Smac—second mitochondria-derived activator of caspases; ICAM1—intercellular adhesion molecule-1.

**Figure 2 metabolites-13-00543-f002:**
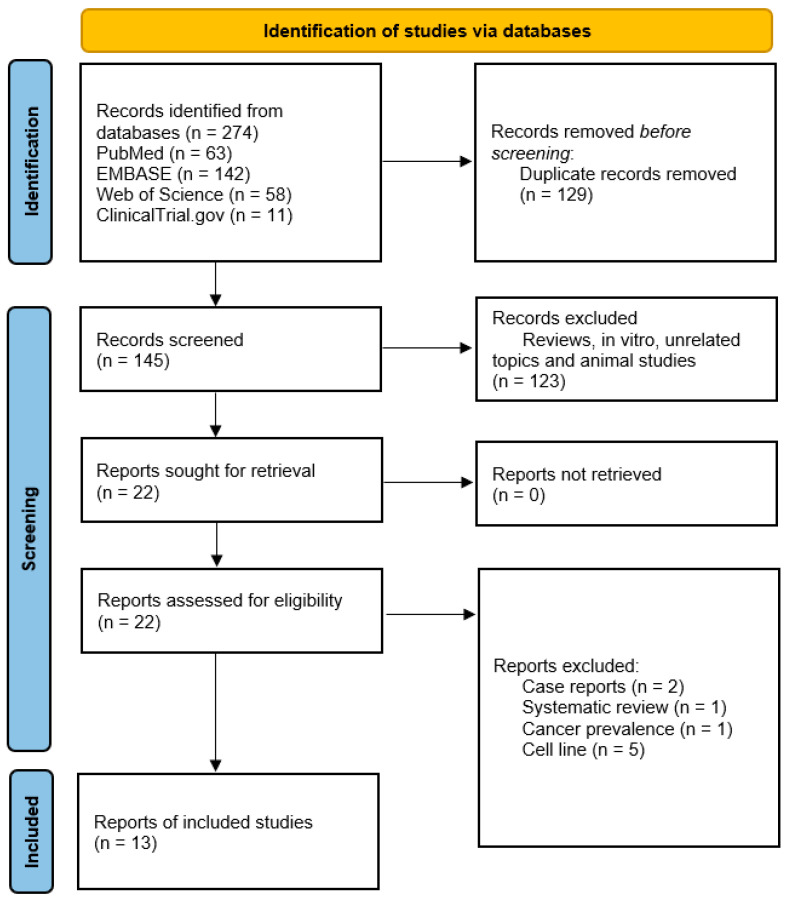
PRISMA chart of the reported studies [119]. We included randomized control and single-arm studies demonstrating thalidomide efficacy in GBM treatment. During screening, we excluded review articles, in vitro studies, unrelated topics, and animal studies.

**Figure 3 metabolites-13-00543-f003:**
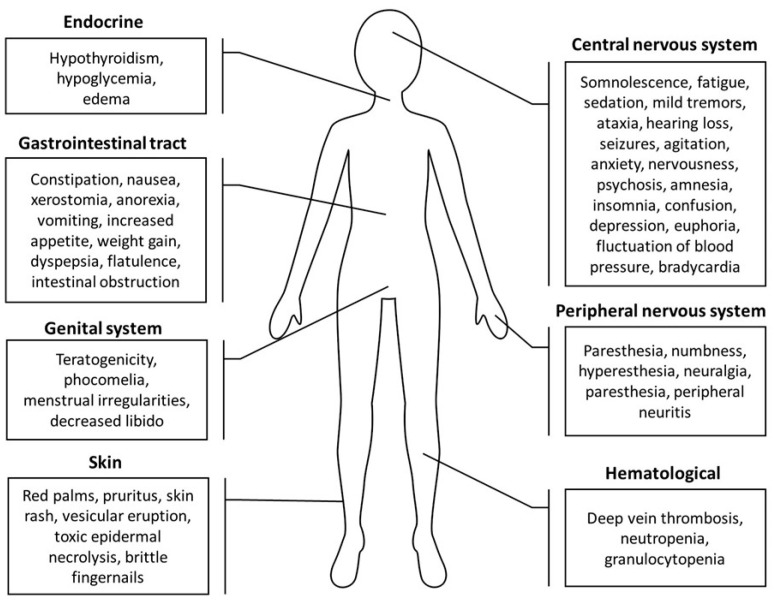
A summary of thalidomide side effects on the skin, gastrointestinal tract, central and peripheral nervous systems, genital system, endocrine system, and hematological system [72,156,158,159].

**Figure 4 metabolites-13-00543-f004:**
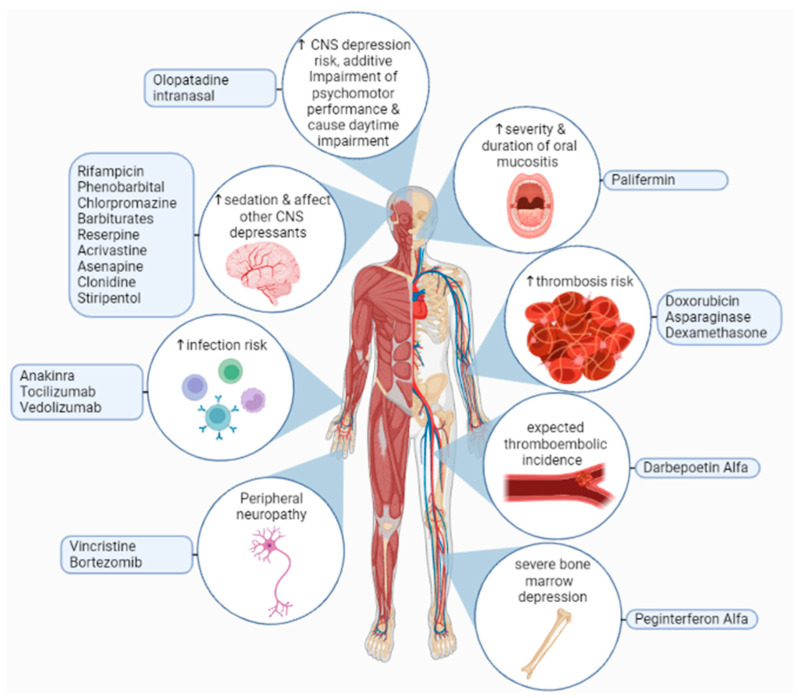
Adverse effects of thalidomide interactions with other drugs may lead to an increased risk of infection, thrombosis, oral mucositis, neurological toxicities, severe bone marrow depression, or increases in the sedative effect of other central nervous system depressants. Furthermore, olopatadine intranasal increases the risk of central nervous system depression, which leads to additive impairment of psychomotor performance, and causes daytime impairment [163,164,165].

**Table 1 metabolites-13-00543-t001:** Thalidomide mechanism of action in vivo.

Disease	Sample Size/Design	Outcomes/Measures	MOA	Ref.
Patients				
Cancer cachexia	416 adult patients with advanced cancer and weight loss or cachexia	Control weight loss in advanced malignancy	Inhibits TNFα and NF-κB, and then limits downstream gene expression, which controls pro-inflammatory cytokines, cells’ growth, and control. Such direct control of NF-κB may help illuminate why THD appears to control severe weight loss with advanced cancers	[109]
Active Crohn’s disease	47 adult patients with active Crohn’s disease received 50 mg THD orally, and then the dose was elevated to 75 mg or 100 mg according to clinical symptoms and tolerance	-Clinical remission at week 8, defined by a CDAI score -Endoscopic efficacy and clinical response	Decreases intestinal inflammatory activity and inflammatory markers (ESR and CRP) and restores mucosal integrity	[117]
Chronic Plaque Psoriasis	20 patients treated with THD at 200 mg/day for eight weeks	THD efficacy in the treatment of psoriasis and in managing the severity of the disease measured by PASI and BSA	Inhibits TNFα	[100]
DLE	60 patients with cutaneous lupus erythematosus, including 25 patients diagnosed with refractory DLE	98% of patients improved after therapy, and 85% attained complete remission over eight years of follow up	-DLE patients had high serum VEGF, THD can decrease IL-10 and TGFβ/Treg production through its immunomodulatory effects on T-cells -THD can inhibit macrophage activation cytokine inhibition, promoting NK cell-mediated cytotoxicity	[98]
ENL	-22 men with ENL received six capsules containing 100 mg of THD (group A, n = 12) Or 300 mg THD (group B, n = 10) daily for a week -Then, group A was given 50 mg/day THD in weeks 2 and 3, then placebo capsules from weeks 4 to 7 -Group B had gradual decrements every two weeks	Resolution of inflamed ENL nodules through the initial seven-day of treatment	Inhibits TNFα associated with ENL toxicity	[94]
HHT	7 HHT patients with recurrent epistaxis were treated with 50 mg/d THD and increased every week by 25 mg/d if no response happened by the third week. The maximum dose was 100 mg/d	THD efficacy for epistaxis that usually occurs in hereditary hemorrhagic telangiectasia	THD could cause blood vessel maturation through increasing expression PDGFRβ in endothelial cells and stimulated activation of the mural cell	[103]
	7 subjects aged between 48 and 75 years had HHT with severe, recurrent epistaxis and mutations in ENG or ACVRL1–3 subjects were untreated controls	THD lowered the frequency of epistaxis in subjects with HHT and reduced blood transfusion requirements	-Antihemorrhagic action through direct inhibition of endothelial cells -THD modulates mural cells activation by increasing their proliferation and forming protrusions that can embrace blood vessels and lead to vessel stabilization	[105]
HIV-1 and Tuberculosis	30 patients were seropositive for HIV-1 infection, never received antiretroviral treatment, and were hospitalized with a recent pulmonary tuberculosis diagnosis	THD effect on HIV levels in vivo and in vitro. -Efficacy of THD on the production of TNFα and Th1-type cytokines (IL-12 and IFNγ) in HIV-infected patients with TB	-Stimulated antigen-specific T cell immunity, as indicated through enhancement of T cell proliferation as a response to PPD -Immunostimulatory effect was indicated by increased plasma levels of some of the Th1-type cytokines and cytokine receptors, including IL-12, IFN-g, TNFαR, and IL-2R -THD increases numbers of CD41 and CD81 T cells and plasma sCD8 levels and stimulates endogenous IL-12 production	[96]
HIV-1	-30 adult antiretroviral naïve males with CD4 ≥ 350 cells/mm^3^ -16 had 200 mg of THD for three weeks; 14 patients in the control group; then, they were followed for 48 weeks	Investigated if THD would decrease HIV replication and the inflammations related to it via measuring CD4/CD8 ratio, CD4 cells count, the percentage of CD38+/HLA-DR+ CD8 cells and US-CRP	Amendment of inflammatory cytokines as IFNγ, TNFα, IL-12, IL-10, COX2, and NF-κB	
IBD	248 patients, including 92 pediatric patients: 192 administered THD, and 56 with Crohn’s disease administered lenalidomide	Induction of remission in pediatric Crohn’s disease	Blocks NF-kB activation after the suppression of IkB kinase activity; THD blocks VEGF and cell adhesion molecule expression in HIMEC	[98]
LVAD patients with GIB	78 patients with LVAD implantation for end-stage heart failure	-Evaluation of efficacy and safety of THD at low dose in the management of LVAD-related GIB	Suppresses VEGF, which implicates arteriovenous malformations	[99]
Multiple myeloma	-84 patients with myeloma, previously treated and progressed -All patients received oral THD for an average of 80 days -The dose started with 200 mg daily, then increased by 200 mg every two weeks until it reached 800 mg per day	Evaluate the efficacy of THD as an antiangiogenic agent in patients with resistant disease	-THD alters the adhesion molecules’ expression and suppresses TNFα production; it increases the production of IL-10 and average soluble IL-2 receptor plasma levels and the total number of lymphocytes, CD4+, and CD8+ T-cell counts -Enhances cell-mediated immunity through cytotoxic T cells’ direct stimulation -Interacts with type 1 and type 2 helper T cells and produces complex effects on levels of cytokines as IL-4, IL-5, and IFNγ	[112]
Multiple myeloma	-499 eighteen years or more patients -They already had received one to three lines of treatment “three different doses of THD (100, 200, or 400 mg/day)” and required additional therapy because of disease progression	-Primary outcome is to compare the time of progression and to know the ideal THD dose considering the time to progression and toxicity. -Secondary ones showed efficacy of THD in the following: -response rate -progression-free survival -overall survival	Unclear	[93]
Neurosarcoidosis	42 years old African American male started THD orally at 400 mg daily for 30-day course	To assess improvement clinically and on MRI	THD is an immunomodulatory drug that inhibits the production of TNFα through enhancing TNFα mRNA degradation; it also produces inhibition of IL-6 upregulation and down-regulates the action of NF-kB, both of which are elevated and essential in CNS inflammation, as in neurosarcoidosis	[102]
Nodular sclerosis type II Hodgkin lymphoma	A 22-year-old woman on 200 mg at night	THD efficacy in controlling severe paraneoplastic pruritus	THD has antipruritic action that can be due to its inhibition of TNFα, or due to its central depressive action	[110]
Oro mucosal disease	-12 patients were given THD through a period of ulcerative oromucosal condition (recurrent aphthous stomatitis, HIV-related oral ulceration, and oral manifestation of Crohn’s disease)	THD had an excellent efficacy-to-safety ratio in managing oro-mucosal ulceration over a prolonged treatment period	Modulation of inflammatory cascade and interaction with various cytokines, such as TNFα, IL-6, and IL-10	[116]
PNP	-12 patients after treatment from associated tumor administered THD (75–100 mg/day) -THD combined with or without low- to mild-dose oral prednisone	-THD could be a safe, effective, and economical treatment choice for PNP patients -The potential of THD in the treatment of PNP through scores of PDAI	-IL-10 and IL-6 were reported to be at raised levels in serum of PNP patients -THD inhibits TNFα, VEGF, IL-6, IL-12, IL-1, IL-10, and IFNγ, and maybe NF-кB -For associated tumors, THD may cause down-regulation of VEGF	[95]
POEMS syndrome	24 adults aged 20 years or more with definite or probable POEMS.	A reduction rate in the concentration of serum VEGF	-THD reduces serum VEGF concentrations and can suppress monoclonal plasma cell proliferation -It causes modulation of up-regulated pro-inflammatory cytokines, such as TNFα, IL6, and IL12	[114]
Portal hypertension secondary to alcoholic cirrhosis	20 alcoholic cirrhosis and current or previously esophageal varices patients	Assessment of THD and oxpentifylline in liver cirrhosis and also their effects on hepatic venous pressure and creation of TNFα	THD reduced hepatic venous pressure gradient through the inhibition of TNFα production	[106]
Pulmonary Tuberculosis	-30 male cases of active TB with either negative or positive HIV1 -They received THD for single or multiple fourteen-day cycles or placebo	-THD did not adversely affect the response of DTH to PPD, differential cell counts, or total leukocytes -Daily intake of THD led to a significant enhancement in weight gain	THD suppresses TNFα production; it increases IFNγ secretion in vivo, in the patient’s serum, and in vitro by PBMCs stimulated by mycobacterium	[113]
PV	6 cases aged between 38 and 67	Appraisal of the safety and efficacy of THD with PV patients	THD inhibits inflammatory cytokines production, including IL-7 and TNFα, and may up-regulate the expression of desmoglein in epidermal keratinocytes as a compensatory mechanism; it also regulates local immunity in the epidermis that contributes to pemphigus pathogenesis	[115]
RAS	-113 patients aged between 18–75 with a history of RAS for about 12 months -The patient had occurrences of aphthae one or two times in one month before enrollment -Fresh minor ulcers presented within two days of eruption without systemic or topical medication treatment -Women refused to become pregnant and started menopause -Men decided to use birth control measures with their partners	-25 mg/d THD extended the recurrence interval of RAS, with better safety through its long-term effect -The primary outcomes were the recurrence interval and total of ulcer-free days -Secondary outcomes were the number of ulcers and the visual analog scale	Unclear	[107]
Animal				
Breast cancer	15 six-week-old female mice	THD’s ability to inhibit the tumor cells and the ability to aggregate and create primary tumors at the injection site	-Delays the capability of a single cell suspension of 4T1 cells to aggregate and create primary tumors -Reduces the density of surface molecules of 4T1 cells implicated in adhesion which would delay the ability of these cells to assemble and create tumors	[111]
Hepatic metastases	20 mice with induced hepatic metastases	Morphological changes in microvessels intratumorally next to THD management in occult hepatic metastases	THD has an antiangiogenic effect that discourages vascular formation, indirectly discouraging tumor progression during the early stages	[108]
HIV-1	-30 adult antiretroviral naïve males with CD4 ≥ 350 cells/mm^3^ -16 have 200 mg of THD for three weeks and 14 patients in the control group; then, they were followed for 48 weeks	Investigated if THD would decrease HIV replication and the inflammations related to it via measuring CD4/CD8 ratio, CD4 cells count, the percentage of CD38+/HLA-DR+ CD8 cells and US-CRP	THD acts as an immunomodulatory, anti-inflammatory, and antiangiogenic drug; it causes the amendment of inflammatory cytokines such as IFNγ, TNFα, IL-12, IL-10, COX2, and NF-κB; THD’s anti-inflammatory effect is exerted by enhancing the degradation of TNFα mRNA	[104]
Spinal cord ischemia/reperfusion injury	54 male rabbits	THD lowered the early-stage ischemia/reperfusion damage of the spinal cord in rabbits	-Inhibits TNFα by enhancing the degradation of TNFα mRNA -THD is a target cell-dependent drug which can target microglia cells; hence, THD can inhibit TNFα production by lipoarabinomannan- or lipopolysaccharide-aroused human microglial cells	[101]
Splenic HSA	15 dogs with a histological diagnosis of HSA and recovery after splenectomy	-The HSA was the primary endpoint for the study -Investigation of the efficacy of THD as an adjuvant drug for canines that diseased by hemangiosarcoma	-THD cannot interrupt the progression of a macroscopic lesion longer -THD can suppress SDF1α, CXCR4, and NF-κB, which can influence the expression of several angiogenesis genes, such as VEGF, β-FGF, and HGF; it can also co-stimulate primary human T lymphocytes, which increases their anticancer activity	[97]

Abbreviations: DLE—discoid lupus erythematosus; Treg—regulatory T; IBD—inflammatory bowel disease; HIMEC—human intestinal microvascular endothelial cells; HHT—hereditary hemorrhagic telangiectasia; LVAD—left ventricular assist device; GIB—gastrointestinal bleeding; ENG—endoglin; ACVRL1—activin A eceptor like type 1; PNP—paraneoplastic pemphigus; has—hemangiosarcoma; IFNγ—interferon gamma; CD—cluster of differentiation; PPD—purified protein derivative; PDAI—pemphigus disease area index; ENL—erythema nodosum leprosum; POEMS—polyneuropathy, organomegaly, endocrinopathy, monoclonal protein, skin changes; RAS—recurrent aphthous stomatitis; DTH—delayed-type hypersensitivity; PBMCs—peripheral blood mononuclear cells; PV—pemphigus vulgaris; IL—interleukin; THD—thalidomide; TNF—tumor necrosis factor; BSA—body surface area.

**Table 2 metabolites-13-00543-t002:** The combined uses of THD with multiple medications.

Combination	Mechanism of Action	Uses	Ref.
-5AZA and THD	Both drugs target the bone marrow microenvironment and cells via different mechanisms. THD has anti-cytokine and antiangiogenic activity, while AZA acts as a prodrug, binds to RNA, and inhibits protein synthesis, affecting the bone marrow microenvironment. After activation of AZA, it binds to DNA, causing cytotoxicity, hypomethylation, gene reactivation, suppression of enzyme methyltransferase, and cell differentiation of MDS	-Useful in treating all MDS types and AML patients with MDS history -In patients with high risks of AML and MDS (n = 40), the efficacy of the combined drug was at least as good as seven days of AZA alone -The survival after one year in continuous combined treatment of AZA and THD is comparable with continuous treatment with AZA	[86]
VDT	The development of therapeutic resistance and cell proliferation in multiple myeloma depends on the bone marrow’s microenvironment. The VDT combination has non-cross resistance, and non-overlapping toxicity as Doxil was a cytotoxic drug that targets the myeloma cells, and THD targets the microenvironment	-Well-tolerated in refractory cases of multiple myeloma (n = 6)	[136]
THD and AZN	In Crohn’s disease, TNFα is implicated in the inflammatory process. AZN has a synergistic impact with THD through its anti-TNFα activity by decreasing the responsiveness of biological treatment and boosting anti-TNFα levels. The synergistic effect of both drugs improves by modulation of T-cell immunity	-Achieving clinical remission (24 weeks, 70.5%) was well-tolerated in Crohn’s disease patients (n = 122) who did not respond to AZN monotherapy -Mucosal healing was achieved in 23.6% of the patients; 50.8% had adverse effects with a 63.6% endoscopic remission rate, and only 13 patients had to discontinue therapy intake	[143]
Capecitabine and THD	THD exerts its immunomodulatory effect, and capecitabine is a prodrug transformed to 5-fluorouracil in tissue via numerous enzymatic routes, and has the same effectiveness in colorectal cancer as 5-fluorouracil	-Treatment of refractory metastatic colorectal cancer (n = 34) revealed a low level of toxicity, such as thromboembolism, constipation, and hand-foot syndrome; however, 38% of the patients had stable disease, and the median progression-free survival rate was 2.6 months, while median overall survival was 7.1 months -There were no radiographic responses observed with therapy -No treatment response in cases treated previously from metastatic colorectal cancer	[145]
Interferon-α and THD	Interferon and THD are angiogenesis inhibitors that slow the progression of renal carcinoma	-The response of renal carcinoma is moderate to interferon and THD -In a study (n = 30) involving patients with renal cell carcinomas, the median follow-up time was 49.6 weeks, while the study’s median time of participation was 11.1 weeks -There were 29 patients with grade 2 toxicity -At 12 weeks, there was no complete response; there were 2 partial response patients, 8 with stable disease, and 11 patients showed disease progression -The median survival was 68 weeks The plasma concentration of THD has no connection to the dose -There was no link between medication level and treatment response or toxic effects	[138]
THD and FCT	Pathophysiology of acute leukemia depends on angiogenesis, contributes to the survival of leukemia cells, and resistance to chemotherapy-induced apoptosis The combination between THD and other chemotherapeutic drugs such as FCT may decrease the angiogenic process due to the ability of THD to block VEGF, thus inhibiting the formation of new vessels	-FCT is an effective therapy in recurrent leukemia patients (n = 42). Adding THD to this combination did not significantly increase efficacy or modify angiogenic markers such as VEGF or MVD -The total response rate in this trial with THD added to the FCT regimen is similar to the regimen seen in phase I when FCT was used alone (24% vs. 26%) -Adding THD results in thrombotic events and dermatological problems, which were not found in phase I using only FCT	[90]
Carboplatin and THD	Carboplatin is an antineoplastic agent that enhances the cell cycle, stimulates apoptosis, and inhibits cancer cell proliferation. Combining both drugs has an antiangiogenic effect through inhibiting cytokines VEGF, TNF, and Factor VIII expression in neoplastic cells	-THD and carboplatin combination effectively inhibit tumor proliferation and 4T1 murine breast cancer metastasis in the animal model -The treated group had a significantly increased animal survival rate compared with the control group (*p* = 0.0005) -Compared with the control group, there was a (62%) significant reduction in tumor growth after treatment (*p* < 0.05) -The number of lung metastases in the treated group was fewer than in the control group (*p* < 0.001) -There was a significant reduction in mitosis and an increase in apoptosis in the treated group compared with the control group (*p* < 0.05 and *p* < 0.001, respectively). -Immunohistochemical analysis of tumor vascularization revealed a higher reduction in the number of blood vessels in the intervention group than in the control (*p* < 0.001)	[74]
TMZ, THD, and celecoxib	TMZ is cytotoxic chemotherapy for the treatment of malignant gliomas. Malignant glioma cells’ secrets of angiogenic factors such as acidic FGF, bFGF, angiogenin, VEGF, platelets-derived growth factors, and IL 8 induce angiogenic process, increasing tumor proliferation. Combining TMZ with two angiogenic inhibitors such as THD and celecoxib inhibits these growth factors, inhibiting angiogenesis	The combination of TMZ, THD, and celecoxib is moderately well-tolerated but unlikely to significantly increase the survival of patients newly diagnosed with GBM (n = 50)	[87]
Bortezomib, THD-dexamethasone	Bortezomib causes inhibition of the action of proteasome, which results in cell apoptosis. Moreover, it inhibits the activation of NF-κB and regulatory protein toxic levels accumulation, but the mechanism leading to apoptosis of cells remains unclear. THD prevents angiogenesis. Finally, dexamethasone suppresses tumor growth and enhances the apoptosis of myeloma cells, but it does not release cytochrome C from mitochondria	-Bortezomib, THD, and dexamethasone combination were successful therapies for multiple myeloma patients (n = 38) -THD and dexamethasone increased induction remission in 66% of patients, but when adding bortezomib to the combination, remission occurred very fast in 87% of patients, including 16% with complete remission -Most side effects of the combination were mild	[88]
Thoracic radiation, paclitaxel, carboplatin, and THD	Inhibition of the angiogenesis process	Combining THD with chemoradiotherapy increased levels of toxicity (grade 3) such as fatigue, dizziness, thromboembolism, tremors, constipation, dyspnea, hypoxia, hypokalemia, rash, edema, sensory neuropathy, and depressed consciousness -Aspirin failed to decrease thromboembolism -THD did not increase survival in people with NSCLC (n = 546), so it was not recommended to add THD to these patients	[141]
THD, vincristine, liposomal doxorubicin, and dexamethasone (T-VAD Doxil)	VAD causes a reduction in tumor cells, especially liposomal doxorubicin, which increases the exposure of tumors cells to doxorubicin. THD works by its anti-myeloma mechanism, and hence, when combined with dexamethasone, it shows synergistic action	The combination was effective in treating myeloma patients (n = 39). Almost 74% of patients responded to treatment; of these, 10% had a complete response, 64% a partial response, 8% a minor response, and 18% did not respond to the treatment -Some patients showed grade 3 or 4 toxicity symptoms such as rash, thrombocytopenia, peripheral neuropathy, constipation, neutropenia, deep venous thrombosis, and 2 patients died due to infections	[85]
THD, IL-2	The effective drug in RCC treatment is the combination immunotherapy drug such as THD with IL-2, which has immunomodulating and antineoplastic activity. Even though the entire mechanism is unknown, the THD/IL-2 combination was found to maintain RCC stability in some patients	-THD and IL-2 were effective in treating metastatic RCC according to phase 1 and phase 2 trials (n = 50), with the benefit ranging from 5–71 percent, with disease stability -The combination was relatively safe, and the only side effects were flu-like symptoms, hypotension, and hypothyroidism, all of which were linked to IL-2 treatment. Constipation, neuropathy, and rash are among the THD toxicity symptoms, and two patients had deep venous thromboses	[89]
Retinoic acid and THD	Retinoic acid induces the activity of resistance genes (HOXB7, bFGF, VEGF, and IL-8) that participate in GBM cell proliferation, hypoxia, and angiogenesis. THD inhibits HOXB7 and bFGF. THD causes suppression of HOXB7 and bFGF. Therefore, the combination shows synergistic action, as THD inhibits the induction of IL-8, IGFBP-3, HILPDA, and ANGPTL4 in GBM tumors, which are elevated by retinoic acid and linked to hypoxia and angiogenesis	The addition of THD to retinoic acid inhibited the development of human U251 GBM xenografts, and suppressed the resistance genes induced by retinoic acid; thus, this combination prevented hypoxia and angiogenesis linked to these genes	[91]
THD and cisplatin	Cisplatin is a toxic agent against murine erythroleukemic cells, and is known for its antimicrobial, immunosuppressive, and mutagenic effects, while THD exerts its effect through its anti-inflammatory and immunomodulatory properties	THD failed to suppress tumor and metastasis when given alone to DBA2/J mice, while cisplatin administration alone or with THD inhibited the multiplication of tumor cells	[150]
Bevacizumab, THD, docetaxel, and prednisone	THD suppresses FGF activity, the proliferation of endothelial cells, circulating endothelial cells, and the appearance of TNF. Bevacizumab keeps the balance of VEGF. Docetaxel has anti-tumor action	The combination helped reduce tumor volume of castration-resistant prostate cancer (n = 60), as the antiangiogenic agents THD and bevacizumab reduced the prostatic specific antigen and achieved an unprecedented response	[148]
Semaxanib and THD	THD down-regulates TNF α, IL-6, VEGF, and FGF. Semaxanib binds to VEGF-2 and causes its inhibition. It also inhibits the growth of A375 melanoma, C6 glioma, Calu-6 lung, and A431 epidermoid xenografts in athymic mice	-The combination of semaxanib and THD is proved to be a successful therapy in malignant melanoma (n = 12) -Headache, asthenia, constipation, edema in the lower limbs, neuropathy, hyperglycemia, hypercholesterolemia, and thrombosis were among the toxicity symptoms encountered by some individuals, whereas headache and thromboembolic symptoms were linked to semaxanib	[147]
THD and infliximab	Infliximab acts as a monoclonal antibody, helps balance TNFα, and down-regulates granulocyte-macrophage colony-stimulating factor expression, which was beneficial in treating TH1 disorders. THD suppresses the formation of TNFα by a mechanism that is different from infliximab	The combination is valuable in the management of refractory entero-Bechet’s disease (n = 1)	[144]
THD and antibiotics (rifampicin and isoniazid)	The anti-inflammatory THD agent enters the CSF and decreases TNFα in CSF and blood. Moreover, it decreases leukocytosis, and its penetration of CSF does not interfere with the action of antimycobacterial drugs	The combination of THD and (rifampicin and isoniazid) improved the survival rate of infected rabbits with mycobacterium tuberculosis	[152]
Pyrotinib and THD	Pyrotinib has been licensed as an effective and irreversible inhibitor of the epidermal growth factor receptor (EGFR)/HER2 identified in NSCLC. THD is effective in solid tumors due to its antiangiogenic and immunomodulatory effects	Since THD lowers diarrhea caused by pyrotinib, a combination of pyrotinib with THD is beneficial in managing NSCLC (n = 39)	[135]
THD and TP chemotherapy	THD and TP chemotherapy reduces VEGF and NRP-1, which are involved in esophageal cancer tumor angiogenesis	Effective in treating esophageal cancer and relieving associated nausea and vomiting (n = 133)	[155]
Glucocorticoids and THD	THD’s immunomodulatory and anti-inflammatory properties allow it to boost T cells, suppress cell growth, and reduce lung fibrosis and damage. THD also calms anxiety and reduces oxygen consumption due to its sedative effect. Glucocorticoids reduce cytokinesis levels	Glucocorticoids and THD in combination effectively relieve symptoms associated with pulmonary effusion without side effects (n = 1)	[137]
THD, IFNα-1b, and IL 2	The combination of IL 2, IFNα-1b, and THD lowers minimal residual disease and reverses AML1-ETO fusion gene positivity in individuals with AML	IFNα-1b, IL-2, and THD are effective in the treatment of AML patients (n = 20), and the effective rate reached 72.2%	[146]
THD and lenalidomide	THD and lenalidomide have direct anti-tumor effects and affect the bone marrow microenvironment: THD has a cytotoxic effect in tumor cells, and lenalidomide inhibits the proliferation of multiple myeloma cells. They induce apoptosis by the down-regulation of NF-kB signaling and enhancing the intrinsic apoptosis pathway. Lenalidomide inhibits the expression of two anti-apoptotic proteins, cIAP-2 and FLICE inhibitory protein. Both proteins block apoptosis induced by TNF-related apoptosis	A broad spectrum of activities to treat multiple myelomas and other hematologic and solid malignancies	[153]
THD, gemcitabine, and carboplatin	THD has an anti-tumor impact on multiple myeloma by inhibiting angiogenesis, mediated by VEGF and bFGF. When coupled with chemotherapeutic drugs, it has a synergistic effect. In mice injected with NSCLC cell lines, THD slows tumor growth	-Overall survival did not improve when THD was combined with chemotherapy -In patients with NSCLC (n = 722), THD was not related to a survival benefit, although survival was lower in the group with non-squamous histology -The most severe side effects of THD were thrombotic events	[142]
THD and irinotecan	THD has immunomodulatory and antiangiogenic activities against malignant gliomas, while irinotecan is a topoisomerase-II suppressor	The combination of THD with irinotecan did not achieve enough effectiveness to warrant further investigation against anaplastic glioma (n = 39) although some patients achieved prolonged progression-free survival rate/overall survival rate	[140],
THD and ciclosporin	THD works similarly to ciclosporin in that it attaches to intracellular proteins, such as calmodulin and cyclophilin, which play a key role in immune response modulation; however, this has yet to be determined. THD interferes with neutrophil and lymphocyte function, causing suppression of the immune system	THD is an effective and potent immunosuppressive drug in preventing early refusal of cardiac allograft transplantation in rats when paired with low dosages of ciclosporin	[139]
THD and intermediate-dose dexamethasone	THD is beneficial in multiple myelomas because it decreases drug resistance in multiple myeloma plasma cells when combined with dexamethasone, so the study used THD and dexamethasone together to treat primary amyloidosis based on these findings	Second-line therapy for amyloidosis (n = 31)	[149]
5-ALA-PDT and THD	5-ALA-PDT is dependent on prodrug 5-ALA administration. It increases protoporphyrin IX in tumor cell production, and acts as a photosensitizer. After radiation therapy, protoporphyrin is excited, leading to ROS formation that causes a cytotoxic effect on tumor cells. Photodynamic therapy enhances VEGF expression and destroys vessels’ walls, leading to a reduction in tumor proliferation. THD inhibits VEGF and TNFα	-This combination was more effective than monotherapy of each agent on the 2H11 endothelial cell line and 4T1 breast carcinomas -In 2H11 cells, THD alone did not affect VEGF expression, but THD lowered VEGF expression once 5-ALA-PDT was administered	[154]
Isotretinoin and THD	Due to P1 or P2 promoter activity, isotretinoin and THD suppress c-MYC expression. Isotretinoin modifies TGFβ expression and also decrease TGFβ signaling degradation of pSMAD1 on the c-MYC promoter. AP-2 and c-MYC interact at the protein level, and AP-2 over-expression reduces c-MYC mRNA expression	-In hepatocellular carcinoma, c-MYC mRNA expression was significantly decreased by 80% (*p* < 0.05) in cells treated with THD and isotretinoin -Isotretinoin effects reduce HepG2 cell viability	[92]
THD and hydroxyurea	The immunomodulating and antiangiogenic effects of THD induce c-globin gene expression, and improve erythroid cell proliferation by amplifying ROS-p38 mitogen-activated protein kinase signaling and histone H4 acetylation during erythropoiesis Hydroxyurea increases hemoglobin by reducing inflammation, hypercoagulability, and induction of HbF expression, which could suppress ineffective erythropoiesis by reducing the build-up and precipitation of precipitation a-globin chains	-For patients with a high risk of organ rejection (n = 25), very high-risk transplant-related mortality and morbidity, or patients who are not willing to undergo a bone marrow transplant -May be used for transplantation in patients with ineligible thalassemia whose iron overload indicates a high risk of treatment-related harm, or who are awaiting donor identification	[151]

Abbreviations: 5-ALA—5-aminolaevulinic acid; ALA-PDT-5—aminolevulinic acid-photodynamic therapy; AML—acute myeloid leukemia; AZA-5—azacytidine; AZN—azathioprine; bFGF—basic fibroblast growth factors; CIAP2—cellular inhibitor of apoptosis protein 2; Cox2—cyclooxygenase 2; DNA—deoxyribonucleic acid; FCT—fludarabine, carboplatin, and topotecan; FGF—fibroblast growth factors; HbF—fetal hemoglobin; IFN—interferon; IL—interleukin; MDS—myelodysplastic syndrome; MVD—marrow microvascular density; NF-κB—nuclear factor-kappa B; NSCLC—non-small-cell lung cancer; RCC—renal cell carcinoma; RNA—ribonucleic acid; ROS—reactive oxygen species; TGF—transforming growth factor; THD—thalidomide.

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
