# Peer review of "Potential Use of Thalidomide in Glioblastoma Treatment: An Updated Brief Overview"

_metabolites, 2023, doi:10.3390/metabo13040543_

Round 1

Reviewer 1 Report

The authors wrote a comprehensive review that highlights the potential benefits of using thalidomide in combination with other medications to treat glioblastoma and its associated inflammatory conditions. They reported also the action mechanism of thalidomide in other tumors.

-Authors have to standardize the names of the genes because sometimes they are in italics and sometimes they are not.

-Authors may report even if HIF2α involvement is known in GBM.

-Authors may report even if pH regulator proteins involvement is known in GBM.

-Authors can add a scheme for Thalidomide drug interaction

Author Response

Please find the attached file with our responses.

Reviewer 2 Report

The authors present an interesting revision regarding the role of Thalidomide in GBM management; the topic is relevant to be discussed since the poor prognosis of grade 4 CNS tumors.

Some methodological and drafting issues should be clarified and corrected before considering this paper for publication:

- Please avoid the definition "multiforme" which is obsolete; furthermore, describe the GBM according to the most recent CNS tumor classification, i.e. the 2021 CNS 5 version: accordingly, GBM should be distinct from astrocytoma IDH-muted grade 4. Please, correct the abbreviation GMB in GBM.

- From a methodological point of view, I suggest the authors to perform a systematic review according to PRISMA guidelines to achieve a greater scientific relevance which can be affected from a selective choice of quoted papers.

- Please extend the section regarding current applications of THD in GBM management and please add a paragraph about further perspectives of this drug in clinical practice, verifying current trial and clinical applications.

Author Response

Please find the attached file with our response

Round 2

Reviewer 2 Report

The authors have improved their study after this process of revision; although their accurate response, I suggest anyway to add a PRISMA diagram of their literature research.

After this further correction, I think that this manuscript can be accepted for publication.

Author Response

Thank you; we added the figure as suggested. We have the figure in the attached file.
